# Biochemical and Structural Characterization of Chi-Class Glutathione Transferases: A Snapshot on the Glutathione Transferase Encoded by *sll0067* Gene in the Cyanobacterium *Synechocystis* sp. Strain PCC 6803

**DOI:** 10.3390/biom12101466

**Published:** 2022-10-13

**Authors:** Eva Mocchetti, Laura Morette, Guillermo Mulliert, Sandrine Mathiot, Benoît Guillot, François Dehez, Franck Chauvat, Corinne Cassier-Chauvat, Céline Brochier-Armanet, Claude Didierjean, Arnaud Hecker

**Affiliations:** 1Université de Lorraine, CNRS, CRM2, F-54000 Nancy, France; 2Université de Lorraine, INRAE, IAM, F-54000 Nancy, France; 3Université de Lorraine, CNRS, LPCT, F-54000 Nancy, France; 4Université Paris-Saclay, CEA, CNRS, Institute for Integrative Biology of the Cell (I2BC), F-91190 Gif-sur-Yvette, France; 5Université de Lyon 1, CNRS, LBBE, F-69622 Villeurbanne, France

**Keywords:** glutathione transferase, glutathione, cyanobacteria, *Synechocystis* sp. PCC 6803, crystallography, biochemistry, phylogeny

## Abstract

Glutathione transferases (GSTs) constitute a widespread superfamily of enzymes notably involved in detoxification processes and/or in specialized metabolism. In the cyanobacterium *Synechocsytis* sp. PCC 6803, SynGSTC1, a chi-class GST (GSTC), is thought to participate in the detoxification process of methylglyoxal, a toxic by-product of cellular metabolism. A comparative genomic analysis showed that GSTCs were present in all orders of cyanobacteria with the exception of the basal order Gloeobacterales. These enzymes were also detected in some marine and freshwater noncyanobacterial bacteria, probably as a result of horizontal gene transfer events. GSTCs were shorter of about 30 residues compared to most cytosolic GSTs and had a well-conserved SRAS motif in the active site (^10^SRAS^13^ in SynGSTC1). The crystal structure of SynGSTC1 in complex with glutathione adopted the canonical GST fold with a very open active site because the α4 and α5 helices were exceptionally short. A transferred multipolar electron-density analysis allowed a fine description of the solved structure. Unexpectedly, Ser10 did not have an electrostatic influence on glutathione as usually observed in serinyl-GSTs. The S10A variant was only slightly less efficient than the wild-type and molecular dynamics simulations suggested that S10 was a stabilizer of the protein backbone rather than an anchor site for glutathione.

## 1. Introduction

Glutathione transferases (GSTs) constitute a widespread superfamily of enzymes playing crucial roles in the cell notably in detoxification processes and in specialized secondary metabolism by catalyzing three major kinds of reactions. These include catalytic reactions where glutathione (GSH) is consumed (GSH-conjugation), reactions where GSH is not consumed (isomerization and dehalogenation) and reactions where GSH is oxidized (thiol-transferase and reduction activities) [1]. At the structural level, canonical GSTs are mainly dimeric proteins, and each subunit adopts a conserved fold composed of an N-terminal thioredoxin (TRX) domain linked to an all α-helical C-terminal domain. The active site of the enzyme is located in a cleft at the interface between both domains and contains a GSH-binding site (G site) and a hydrophobic-substrate binding site (H site). Depending on their primary sequence conservation, GSTs were classified into classes designated by a Greek letter. GSTs with a sequence identity greater than 40% belong to the same class, whereas proteins of different classes share less than 25% sequence identity [2]. GSTs were further distinguished into four catalytic types, tyrosine (TyrGSTs), serine (SerGSTs), cysteine (CysGSTs) and atypical (AtyGSTs), depending on an assumed important residue for catalysis [3]. The tyrosine, serine and cysteine residues have a conserved position in the structures. The tyrosine residue is located at the C-terminal end of the first strand (β1). The serine and cysteine residues have the same position at the N-terminal end of the first helix (α1). AtyGSTs do not have a specific residue at a conserved position. In CysGSTs, the cysteine residue has a reactant role according to the enzyme mechanism ontology [4], since it forms a covalent bond with the substrate during the catalytic act. The important residue of the others (TyrGSTs, SerGSTs and AtyGSTs) has a spectator role in enhancing the nucleophilicity of the glutathione thiolate group, and the mutation of this residue does not abolish the activity. Through noncatalytic properties, so-called ligandins, hitherto underestimated compared to the other documented roles of GSTs, many GSTs also participate in the binding and transport of small heterocyclic ligands [5,6].

GSTs have been extensively investigated in animals and plants because of their great relevance to human health and agriculture [7,8,9]. In contrast, studies in bacteria remain scarce, especially in the cyanobacterial phylum that encompasses oxygenic photosynthetic prokaryotes considered to be the ancestors of chloroplasts. It has been speculated that cyanobacteria may be the first organisms to harbored GSTs [10]. Greek letters have been used for eight classes of GSTs from bacteria: beta, chi, eta, nu, rho, theta, xi and zeta [11,12]. Other classes exist that often have specific functions such as LigE, LigF and LigG involved in lignin degradation in soil bacteria [13,14].

The chi class is thought to be specific of cyanobacteria and three isoforms (TeGSTC1 from *Thermosynechococcus elongatus* BP-1, SeGSTC1 from *Synechococcus elongatus* PCC 6301 and SynGSTC1 from *Synechocystis* sp. PCC 6803) have been characterized biochemically [10,15,16]. A preliminary crystallographic study has been reported for TeGSTC1 and SeGSTC1 [17]. All the three isoforms (TeGSTC1, SeGSTC1 and SynGSTC1) exhibit similar activities. They efficiently catalyze the addition of GSH on various isothiocyanates and show moderate activities toward other classical substrates such as chlorodinitrobenzene [10,15,16]. Interestingly, we recently showed that SynGSTC1 is involved in the detoxification of methylglyoxal, a toxic by-product of the cellular metabolism [15]. Despite a shorter sequence length compared to other GSTs, homology modelling combined with secondary structure prediction suggested that chi GSTs (GSTCs) adopt the fold of canonical GSTs. Their amino acid sequences show two motifs usually found in GSTs. The motif I, which contains an invariant cis-proline residue as well as a ββα structure essential for the stabilization of the γ-glutamyl moiety of GSH, is the most conserved region in all of the GSTs [18]. Motif II, in turn, contains a very well conserved aspartic acid important for fold stability [19]. Recently, a conserved tyrosine residue located at the fifth position of the N-terminus of GSTCs has been proposed as the catalytic residue [20]. To better characterize the chi class of GSTs, it was necessary to obtain an experimental three-dimensional model. Therefore, we determined the first crystal structure of a chi-class GST (SynGSTC1), performed a robust phylogenetic study and completed the biochemical data by testing new substrates and modulating the active site residues by site-directed mutagenesis.

## 2. Materials and Methods

### 2.1. Cloning, Mutagenesis, Expression and Purification

SynGSTC1 (Sll0067) encoding sequence was amplified by PCR from *Synechocystis* sp. PCC 6803 genomic DNA as template using specific forward and reverse primers containing *Nde*I and *Xho*I restriction sites, respectively (Appendix A). The amplified sequence was subsequently digested and cloned in *E. coli* expression vector pET-26b between *Nde*I and *Xho*I restriction sites allowing the fusion of a His-tag at the C-terminal part of SynGSTC1 as previously described [15]. Various catalytic mutants (S10T, S10A, S10C and R11A) were generated by site-directed mutagenesis using the QuikChange site-directed mutagenesis kit (Agilent Technologies) and specific mutagenic primers listed in Appendix A. The sequences have been confirmed by DNA sequencing.

The expression of recombinant SynGSTC1 and variants were performed at 37 °C using *E. coli* Rosetta2 (DE3) pLysS expression strain (Novagen) transformed with appropriate plasmid in LB medium supplemented with kanamycin (50 μg/mL) and chloramphenicol (34 μg/mL). When the cell culture reached an OD_600 nm_ of 0.7–0.8, the expression of the SynGSTC1 (or S10T or S10A or S10C or R11A) recombinant protein was induced with 0.1 mM isopropyl β-D-1-thio-galactopyranoside (IPTG) for 4 h at 37 °C. Cells were then harvested by centrifugation, resuspended in a 30 mM Tris-HCl buffer (pH 8.0) supplemented with 200 mM NaCl (lysis buffer) and stored at −20 °C until use. After the lysis of the cells by sonication, the resulting cell extract was centrifuged at 40,000× *g* for 20 min at 4 °C to remove cellular debris and aggregated proteins. After the addition of 10 mM imidazole, SynGSTC1 was purified from the soluble extract by gravity-flow chromatography on a nickel nitrilotriacetate (Ni-NTA) agarose resin (Qiagen, Hilden Germany). After a washing step with lysis buffer containing 20 mM imidazole, recombinant SynGSTC1 was eluted using lysis buffer supplemented with 250 mM imidazole. The fractions of interest were pooled, concentrated by ultrafiltration, subjected to a size exclusion chromatography using a Superdex^TM^200 16/600 column connected to an ÄKTA-Purifier^TM^ device (Cytiva) and eluted with lysis buffer. The purified recombinant protein was concentrated and finally stored at −20 °C. The concentration of SynGSTC1 recombinant protein was determined at 280 nm using a theoretical molar absorption coefficient of 28,420 M^−1^·cm^−1^.

### 2.2. Crystallization, X-ray Data Collection, Processing and Refinement

A first screening of 288 crystallization conditions was carried out at the CRM2 crystallogenesis platform (University of Lorraine) with an Oryx 8 crystallogenesis robot (Douglas Instruments Ltd, Hungerford, UK). Crystals were optimized manually at 4 °C by the microbatch-under-oil method. Solutions of SynGSTC1 ant the variants contained 30–40 mg·mL^−1^ protein in 30 mM Tris buffer (pH 8.0) supplemented with 200 mM NaCl, 1 mM Tris(2-carboxyethyl)phosphine (TCEP) and 10 mM glutathione. SynGSTC1 was crystallized by mixing 1 µL of protein with 1 µL of solution consisting of 16% (*w/v*) PEG 8000, 40 mM potassium phosphate monobasic and 20% (*w/v*) glycerol (condition no. 32, Wizard™ Classic Crystallization Screen III, Rigaku, Tokyo, Japan).

Preliminary X-ray diffraction experiments were carried out in-house on an Agilent SuperNova diffractometer (Rigaku Oxford Diffraction) equipped with a CCD detector. Data collections were carried out at the ESRF, on beamline FIP BM07 (ESRF, Grenoble, France) and (PX1 and PX2, SOLEIL, Gif-Sur-Yvette, France). Data sets were indexed and integrated with XDS [21], and scaled and merged with Aimless [22] from the CCP4 suite [23]. The structure of SynGSTC1 was solved by molecular replacement using *MoRDa* [24] with the coordinates of a GST from *Rhodobacter sphaeroides* (PDB entry 3LSZ) as the search model. Structures of SynGSTC1 and its variants were refined with *BUSTER* [25] and manually improved with *Coot* [26]. The validation of all structures was performed with the PDB validation service (http://validate.wwpdb.org, accessed on 30 September 2022). The coordinates and structure factors have been deposited in the Protein Data Bank (PDB entries 8AI8, 8AI9, 8AIB). Crystal data, diffraction and refinement statistics are shown in Table 1.

### 2.3. Structure Analysis Based on Electron Density Distribution

To calculate the electrostatic interaction energies between residues of SynGSTC1 active site and the glutathione ligand, the electron charge density of the complex based on the Hansen and Coppens multipolar model [27] was determined (method detailed in Appendix A). The electron density parameters for the SynGSTC1-GSH complex were transferred from the ELMAM2 database2, which provides parameters averaged over experimental peptide electron densities from ultra-high resolution X-ray scattering data [28]. In addition, polarization effects due to the environment were estimated in the transferred electron density using the procedure described recently by Leduc et al. [29] and implemented in *MoProViewer* software (version 0.1.1302) [30]. The electrostatic interaction energy ($E_{tot}^{elec}$) between the glutathione ligand and the SynGSTC1 active site residues was computed using *Charger*, which is a fast and analytical electrostatic energy calculation tool [31] also implemented in *MoProViewer*. $E_{tot}^{elec}$ includes two terms, the electrostatic interaction permanent energy $E_{perm}^{elec}$ and the polarization contribution $E_{pol}^{elec}$ (hence $E_{tot}^{elec}=E_{perm}^{elec}+E_{pol}^{elec}$). The *MoProViewer* database transfer tool enables an automatic parameter transfer on the structure with appropriate formal charge assignment (+1e for arginine and lysine, −1e for aspartate and glutamate, 0 for others). The His38 and His61 of SynGSTC1 were protonated on the Nε atom and the formal charge of glutathione was set to −1e. The procedure is detailed in the Appendix A. 

### 2.4. Molecular Dynamics Simulation

The Molecular Dynamics simulations presented in this study were based on the crystallographic structure of SynGSTC1 in complex with GSH. This system was immersed in a cubic simulation cell of length equal to 73 Å filled by a solvent of 9881 water molecules with 150 mM NaCl. The simulations were performed using *NAMD 3.0* [32] with the CHARMM36 [33] force field for proteins and the TIP3P water model [34]. The parameters for the GSH ligand were generated by the CHARMM general force field (CGenFF) [35]. Long-range electrostatic forces were evaluated using the particle mesh Ewald algorithm with a grid spacing of 1.0 Å. A smoothed 12.0 Å spherical cutoff was applied to truncate the short-range van der Waals and electrostatic interactions. The temperature was maintained at 300 K thanks to the Langevin thermostat and the pressure at 1 atm thanks to the Langevin piston method. Covalent bonds involving hydrogen atoms were restrained to their equilibrium length by the Rattle algorithm [36] and the water molecules were constrained to their equilibrium geometry using the Settle algorithm [37]. In addition, a mass-repartitioning scheme was used to integrate the equations of motion with a time step of 4 fs, according to Hopkins et al. [38]. A smooth equilibration, along which the positions of the heavy atoms of the protein were restrained harmonically, was carried out during 8 ns before a non-restrained long equilibration of 100 ns. Then, the SynGSTC1-GSH complex was probed in production runs including a long simulation of 500 ns and five independent shorter simulations of 100 ns. These trajectories were visualized and analyzed using *VMD* [39]. These simulations were aimed at exploring the stability of the interactions between the glutathione and the active site as well as the flexibility of the protein interdomain linker.

### 2.5. Enzymatic Assays

The GSH-conjugation activity was assayed at 25 °C toward 1-chloro-2,4-dinitrobenzene (CDNB), benzyl-isothiocyanate (BITC), 2-phenetyl-isothiocyanate (PITC) or p-nitrophenyl butyrate (PNP-butyrate). The reactions were performed in 500 μL of 30 mM Tris-HCl (pH 8.0) and 1 mM EDTA for CDNB and PNP-butyrate and 100 mM phosphate buffer (pH 6.5) for ITC derivatives in the presence of various concentrations of CDNB (0–4000 µM), BITC (0–1000 µM), PITC (0–1000 µM) or PNP-butyrate (0–2000 µM) at a fixed saturating GSH concentration. Peroxidase and thiol-transferase activities were assayed at 25 °C toward cumene hydroperoxide (CuOOH) and 2-hydroxyethyl disulfide (HED) in a NADPH-coupled spectrophotometric method by following the absorbance at 340 nm. The reactions were carried out in 500 μL of 30 mM Tris-HCl (pH 8.0) containing 200 μM NADPH, 0.5 unit of yeast glutathione reductase and various concentrations of HED (0–500 µM) or CuOOH (0–3000 µM) at a fixed GSH concentration. The optimum pH of the wild-type enzyme and its variants was determined against PITC using 100 mM sodium citrate, phosphate, or borate buffers at pH ranging from 4.0 to 11.0. GSH-conjugation activity was determined as described above.

For all activity assays, the recombinant protein, used at a concentration (3 µM) within the linear response range of the enzyme, was added after 2 min of preincubation and the variation of absorbance monitored using a Cary 50 spectrophotometer. The activity recorded without enzymes was subtracted and three independent experiments were performed at each substrate concentration. The kinetic parameters, apparent *K*_m_ (Michaelis constant) and *ksoftware*_cat_ (turnover number) were determined by fitting the data to the nonlinear regression Michaelis–Menten model in *GraphPad Prism* (version 8, GraphPad Software, Inc., San Diego, CA, USA). The *k*_cat_ values were expressed as μmol of substrate oxidized per second per μmol of enzyme (i.e., the turnover number in s^−1^) using specific molar absorption coefficients of 9600 M^−1^·cm^−1^ at 340 nm for CDNB, 9250 M^−1^·cm^−1^ at 274 nm for BITC, 8890 M^−1^·cm^−1^ at 274 nm for PITC, 17700 M^−1^·cm^−1^ at 412 nm for PNP-butyrate and 6220 M^−1^·cm^−1^ at 340 nm for NADPH.

### 2.6. Phylogenetic Analysis

In total, 222 proteomes of the Cyanobacteria/Melainabacteria group were retrieved from the RefSeq database of the NCBI. These corresponded to 208 proteomes of Cyanobacteria labelled as RefSeq “reference proteomes” or from type strains, and 14 proteomes from noncyanobacterial lineages (i.e., Margulisbacteria, Melainabacteria, Gastranaerophilales) classified in the Cyanobacteria/Melainabacteria group (Appendix A). The sequences of the 53 ribosomal protein families (rprots) were retrieved from the 222 proteomes using the riboDB database [40] (Appendix A). The corresponding protein sequences were aligned using *MAFFT v7.453* with the accurate option L-INS-I [41]. The resulting multiple alignments were trimmed with *BMGE v1.2* using the BLOSUM30 substitution matrix [42]. The multiple alignments of the 52 rprots present in more than 30% of the 222 analyzed proteomes were combined to build a large supermatrix (222 sequences, 6430 amino acid positions) and used to build a phylogeny using the maximum likelihood method. The tree was inferred with *IQ-TREE* (multicore version 2.2.0 COVID-edition, June 2022) with the LG + C20 + F + R4 evolutionary model [43]. The branch robustness of the inferred tree was computed with the ultrafast bootstrap procedure implemented in *IQ-TREE* (1000 replicates). The resulting tree was rooted using the 14 noncyanobacterial sequences. 

The 222 studied proteomes were queried with *BLASTP* using the GSTC1 sequence from the *Synechocystis* sp. PCC 6803 strain (RefSeq protein Id WP_010873500.1, locus tag SGL_RS13850) as seed. The 924 GST sequences displaying an E-value lower than 10^−3^ were retrieved and aligned using *MAFFT* with the auto option. A total of 54 partial sequences were discarded from the analysis. A survey of the nr database at the NCBI identified 11 sequences of GSTC in noncyanobacterial bacteria. These 11 sequences were added to the cyanobacterial GSTC sequences. The 881 GSTC sequences were realigned with *MAFFT* with the L-INS-I option and trimmed using *BMGE* with the BLOSUM30 substitution matrix. The 104 kept amino acid positions were used to infer a phylogeny using *FastTree v2* [44] with 20 rate categories of sites, the gamma optimization option, and the Le and Gascuel model [45]. The branch robustness of the inferred tree was estimated using the Shimodaira Hasegawa test (resampling the site likelihoods 1000 times). Finally, a phylogenetic analysis of the 147 cyanobacterial GSTC sequences was performed using FastTree and the same parameters (147 sequences, 110 amino acid positions).

The trees were drawn using *iToL v6.5.8* [46].

## 3. Results and Discussion

### 3.1. Crystal Structure of SynGSTC1

In this study, the crystal structure of the glutathione transferase chi1 from *Synechocystis* sp. PCC 6803 (SynGSTC1) in complex with GSH is presented. We also solved the structures of two variants (S10T and R11A variants in complex with GSH) which did not show significant differences from the wild-type. The protein samples were cocrystallized with an excess of GSH (10:1) in the presence of TCEP to avoid oxidation of the GSH thiol group into sulfenic acid. SynGSTC1 crystallized in space group *P*4_3_2_1_2 with two polypeptide chains in the asymmetric unit. They formed a two-fold dimer that had a globular shape with molecular dimensions of approximately 55 Å× 55 Å × 45 Å (Figure 1). The dimer buried 1710 Å^2^ of surface area for each monomer and was tightly stabilized by ten hydrogen bonds and six salt bridges (Appendix A). At the core, a four-helix bundle consisting of the α3 and α4 helices of the two monomers buried aliphatic residues (L70 and L94 of chains A and B). This interaction pattern was complemented by a lock-and-key motif where the F49 residue fitted into a low-polar cavity of the adjacent subunit (W92, F95, L117, L121) (Figure 1).

Both subunits were very similar structures and could be superimposed within 0.33 Å root-mean-square deviation over 181 α-carbon atoms. The SynGSTC1 protomer adopted the conserved GST fold that was subdivided into two domains for clarity (N-terminal domain β1α1β2α2β3β4α3 and C-terminal domain α4α5α6α7α8). As mentioned in the introduction, the chain length of GSTCs (approximately 180 residues) was significantly shorter by at least 20 residues compared to most canonical GSTs [47]. The α4–α5 hairpin pattern was significantly shortened (roughly 10 residues) and the angle between these two helices (~42°) was twice that usually observed (Figure 2). This “missing” region made the active site of SynGSTC1 very open, with no clear pocket for the hydrophobic substrate (H-site). Both motifs I (47–71) and II (129–147) played their expected structural roles. In motif I, the V52–P53 peptide bond was cis, and V52 formed the typical antiparallel β-sheet-like interaction with the cysteine moiety of GSH. Motif II contained the Ncap sequence ^137^SVVD^140^ where the side chains of the serine and aspartic acid residues contributed to the stabilization of the α6 helix [19]. The linker (^76^ASTIPAD^82^) between the N- and C-terminal domains was peculiar because it had no aliphatic or aromatic residue wedged between these two domains as usually observed [48,49]. The consequence was an interdomain linker without a unique conformation. The quality of the electron density allowed the building of two major conformations (Figure 1 and Appendix A). To investigate this property, we performed molecular dynamics simulations of the SynGSTC1-GSH complex in an aqueous environment. The simulation revealed a protein very stable with the linker as one of the most mobile regions. The time-evolution of the φ and ψ torsion angles of the linker residues revealed transitions between two main conformations during the trajectory (Appendix A). Interestingly, these two conformations corresponded to those observed in the crystal structure.

### 3.2. Structural Comparison

A search for the structural homologs using the Dali server (http://ekhidna2.biocenter.helsinki.fi/dali/, accessed on 30 September 2022) ranked proteobacterial nu GSTs and fungal GSTs from the Ure2p class at the top of the list [50]. The other hits included proteobacterial beta GSTs, insect delta GSTs, an unclassified proteobacterial GST and plant phi GSTs. To better depict the proximities of these structures, an additional multiple structural alignment was performed using the mTM-align server (https://yanglab.nankai.edu.cn/mTM-align/, accessed on 30 September 2022) [51] (Figure 2). The resulting dendrogram based on the pairwise alignment scores (Appendix A) showed a distribution of the proteins into two clades, one of which containing SynGSTC1 and the unclassified proteobacterial GST (GST SMc00097 from *Sinorhizobium meliloti* 2011, PDB entry 4nhw). The latter had therefore the most similar structure to SynGSTC1. SMc00097 had one of the structural attributes of SynGSTC1, namely a SRAS motif at the beginning of the α1 helix in its active site (see below) (Figure 2). The first serine residue adopted the same orientation and did not participate in the stabilization of GSH while the arginine residue did (Appendix A). The closeness between SynGSTC1 and SMc00097 could be explained in a more comprehensive way by a domain-by-domain comparison. Indeed, the overall structures (i.e., both the N-ter and C-ter domains) of SynGSTC1 and SMc0097 overlapped well (Appendix A). The proximity of SynGSTC1 with other hits (nu, beta, delta and phi GSTs) was rather due to the good overlap of N-terminal domains. 

### 3.3. Active Site Structure and Its Analysis Using Transferred Multipolar Electron-Density

The active site contained GSH tightly bound to the G-site by numerous polar interactions (respectively, six, two and three for the γ-Glu, Cys and Gly moieties) (Figure 3). The GSH Cys moiety adopted two rotamers (**m**, χ_1_ = −52° and **t**, χ_1_ = 172°) exposing the GSH thiol group towards the solvent (Figure 1). The three regular rotamers (**p**, **m**, and **t**) of the glutathione cysteine moiety were accessible during the molecular dynamics simulations with the frequencies of 0.35, 0.42 and 0.16, respectively (Appendix A). The crystal structure did not reveal a strong polar interaction between the sulfur atom of GSH and the enzyme. The smallest distance was 3.8 Å with the amide group of R11. The Y5 residue, recently proposed as a catalytic residue [20], was far too distant to stabilize the GSH-thiolate group during catalysis as the Y5 hydroxyl group was 17 Å away from the GSH sulfur-atom. Based on the sequence analysis of SynGSTC1, we could have thought that S10 played an important role in catalysis. Indeed, this serine residue belongs to the ^10^SRAS^13^ motif, which is related to the CXXC active-site motif of thioredoxin [52]. The equivalent serine residue in Ser-GSTs (see introduction) is almost always found hydrogen-bonded to the GSH thiol group while this is not the case in SynGSTC1 [53,54]. Indeed, the OG atom invariably retained the same orientation in all structures (wild-type and variants), and was hydrogen bonded to the main chains of A7 and A12. This interaction network remained stable throughout most of the molecular dynamics simulations showing that S10 was important for the stabilization of the protein backbone. Whatever its conformation, this serine residue never formed a strong interaction with the GSH thiol group during the simulation (Appendix A).

The description of the interactions between a ligand and a protein is most often summarized by the list of residues involved, without quantifying the importance of each. We developed recently a fast and analytical procedure to estimate the electrostatic contribution of each residue to the ligand binding, based on a continuous distribution of electron density of experimental origin [31]. This method implemented in *MoProViewer* [30] was applied on SynGSTC1 in complex with GSH where the contributions of eleven residues were evaluated (distance cutoff of 3.5 Å away from GSH). This included eight residues from one chain (S10, R11, L33, H38, K51, V52, E64, S65 and N97) and three from the other (S98, T99 and R116). *MoproSuite* calculates electrostatic interaction energies $E_{\mathrm{tot}}^{\mathrm{elec}}$ that are divided into two contributions: a permanent electrostatic interaction term, $E_{\mathrm{perm}}^{\mathrm{elec}}$, and a polarization one, $E_{\mathrm{pol}}^{\mathrm{elec}}$, which can be interpreted as a molecular recognition term and an adaptation term, respectively (Figure 4, Appendix A). By definition, the polarization term is negative and makes the total interaction energy more favorable for all the active site residues and especially for charged residues [29]. We performed the calculations for the two GSH thiol orientations observed in the crystal structure. The orientation of the thiol group did not affect notably the GSH binding, from an electrostatic and dipolar-induction point of view (Appendix A). Thus, the following analysis did not depend on the GSH conformation.

The permanent interaction energy $E_{\mathrm{perm}}^{\mathrm{elec}}$ pictures the electrostatic complementarity between the GSH chemical groups and the residues lining the binding site. GSH was assumed to bear three charges: a zwitterionic γ-glutamic acid moiety and a terminal glycine carboxylate group. The SynGSTC1 residues with the largest contributions were R11, K51, R116, which formed salt bridges with the GSH negative charges (Figure 3, Appendix A). As an example, the energy values $E_{\mathrm{tot}}^{\mathrm{elec}}$, $E_{\mathrm{perm}}^{\mathrm{elec}}$ and $E_{\mathrm{pol}}^{\mathrm{elec}}$ obtained for R11 were −56.6 kcal·mol^−1^, −47.3 kcal·mol^−1^ and −9.3 kcal·mol^−1^, respectively. S65 showed the most favorable $E_{\mathrm{perm}}^{\mathrm{elec}}$ among the uncharged residues ($E_{\mathrm{perm}}^{\mathrm{elec}}=-39.3\;\mathrm{kcal}\cdot {\mathrm{mol}}^{-1}$), and its contribution was close to those of R11 and R116 when the dipolar induction is included ($E_{\mathrm{tot}}^{\mathrm{elec}}=-54.3\;\mathrm{kcal}\cdot {\mathrm{mol}}^{-1}$). This serine residue was double-hydrogen-bonded to the γ-Glu carboxylate group. This interaction pattern, well conserved in GSTs, is ensured either by a serine residue or a threonine residue [55]. The negatively charged E64 residue was an interesting case because it had an unfavorable $E_{\mathrm{perm}}^{\mathrm{elec}}$ ($13.6\;\mathrm{kcal}\cdot {\mathrm{mol}}^{-1}$) that underwent a significant dipolar induction ($E_{\mathrm{pol}}^{\mathrm{elec}}=-19.7\;\mathrm{kcal}\cdot {\mathrm{mol}}^{-1}$) to interact with the positively charged N-terminal amine group of GSH ($E_{\mathrm{tot}}^{\mathrm{elec}}=-6.0\;\mathrm{kcal}\cdot {\mathrm{mol}}^{-1}$, Figure 4, Appendix A). The major contributors for the GSH γ-Glu moiety, $E_{\mathrm{perm}}^{\mathrm{elec}}$ speaking, were therefore R11 via its guanidium group and S65 via its amide and hydroxyl groups (Figure 3). This showed that the site where the zwitterionic fragment of GSH was located was an electrophilic site. This property is verified in the crystallographic structures of glutathione-free GSTs because they often contain a negative ion in this site such as chloride, acetate or formate ions [56]. In addition, this electrophilic site was found to be catalytically important as it hosts the γ-Glu carboxylate group which is presumed to decrease the pKa of the GSH thiol group [57]. The glycine part of GSH was surrounded by the two positively charged K51 and R116 residues, and by the lateral chain of H38 residue. These residues tightly stabilized the GSH C-terminal carboxylate group (Figure 3 and Figure 4). Finally, the GSH Cys part was strongly stabilized by a single residue (V52) via two main-chain–main-chain hydrogen bonds (Figure 3). This twofold contribution was significantly lower compared to that of S65 probably because the V52-GSH interaction did not involve charged groups. The S10 residue, which was assumed to be the catalytic residue interacting with the thiol group, presented an unfavorable electrostatic interaction energy and did not contribute to the GSH stabilization ($E_{\mathrm{tot}}^{\mathrm{elec}}=1.3\;\mathrm{kcal}\cdot {\mathrm{mol}}^{-1}$, Appendix A). It also showed an almost zero polarization energy so this residue was not affected by the binding of the glutathione. This correlated well with the fact that the crystal structure of SynGSTC1 revealed no intermolecular interaction between S10 and GSH. The side chain of the “main” tyrosine residue of TyrGSTs (Tyrosine type GSTs) was always observed interacting with the thiol group of GSH in the crystal structures. The “main” serine residue of SerGSTs plays the same role in most known structures. We evaluated the electrostatic contribution of residues to GSH binding in a TyrGST (and a SerGST) containing a putative hydrogen bond between the tyrosine (serine) residue and GSH (Appendix A). In both cases, the important residue (tyrosine or serine) provided a stabilizing effect on the ligand ($E_{\mathrm{tot}}^{\mathrm{elec}}=-12\;\mathrm{kcal}\cdot {\mathrm{mol}}^{-1}$ and $E_{\mathrm{tot}}^{\mathrm{elec}}=-7.3\;\mathrm{kcal}\cdot {\mathrm{mol}}^{-1}$, respectively, Appendix A). However, this contribution was never predominant. The main anchor points remained the positively charged residues that stabilized the terminal carboxylate groups of GSH. 

### 3.4. Biochemical Characterization of SynGSTC1 and Variants

We recently detected an activity for SynGSTC1 toward methylglyoxal as substrate and also tested glutathione transferase reactions namely aromatic substitution, and addition using, respectively, 1-chloro-2,4-dinitrobenzene (CDNB) and isothiocyanates (ITCs) as substrates [15]. In addition to these activities, we tested here the ability of SynGSTC1 to conjugate GSH on 4-nitrophenyl butyrate (PNP-butyrate) by transacylation and to reduce hydroperoxide toward cumene hydroperoxide (CuOOH). The measured activities (*k*_cat_/*K*_m_), respectively of 49.0 ± 1.7 M^−1^.s^−1^ for PNP-butyrate and 604.6 ± 37.7 M^−1^·s^−1^ for CuOOH are similar to the one measured toward CDNB 112.5 ± 14.2 M^−1^·s^−1^. These activities remain significantly lower than those measured with ITCs (6.7 × 10^5^ ± 0.2 × 10^5^ M^−1^·s^−1^ and 5.7 × 10^5^ ± 0.2 × 10^5^ M^−1^·s^−1^ for PITC and BITC, respectively) due to a higher affinity of the enzyme for PITC and BITC (31.4 ± 3.5 and 82.0 ± 10.0 µM, respectively) associated to a higher turn-over number (21.0 ± 0.5 s^−1^ and 45.0 ± 1.6 s^−1^, respectively) (Appendix A).

We also investigated the structure–activity relationships of SynGSTC1 by targeting the first two residues of the ^10^SRAS^13^ motif, S10 being suspected to activate glutathione as in Ser-GSTs and R11 because of its ubiquity in GSTCs (see below in Section 3.5). The kinetic constants and the effect of pH on activities toward PITC were determined for S10T, S10A, S10C and R11A variants (Table 2 and Appendix A). The optimal pH of SynGSTC1 WT (7.4 units) was in the same range as usually observed for GSTs [57]. The substitution of S10 by a threonine residue slightly decreased the optimal pH of the enzyme (6.9 vs. 7.4 for WT) and the catalytic efficiency (*k*_cat_/*K*_m_) of the protein (2.1 × 10^5^ M^−1^·s^−1^ for S10T vs. 3.7 × 10^5^ M^−1^·s^−1^ for WT). This result was consistent with the crystal structure of S10T which was superimposable to the wild-type (Appendix A). The bulkier threonine side chain did not impair GSH binding. Indeed, the GSH apparent affinity (*K*_m_) was not altered in the S10T variant (Table 2). Furthermore, a sequence analysis of GSTCs (see below in Section 3.5) showed either a serine or a threonine as the first residue of the active site motif (^10^SRAS^13^ in SynGSTC1). S10A remained effective even though the decrease was greater than for S10T, being divided by 10 and 1.2 in S10A and S10T, respectively, compared to WT. All the kinetic parameters were affected roughly similarly. The crystal structure of SynGSTC1 did not show interaction between S10 and GSH. Instead, S10 was rather involved in stabilizing the β1-α1 loop in the close vicinity of the G-site (see above in Section 3.3) suggesting that the S10A substitution most likely disrupted the integrity of the active site. This resulted in a degradation of the catalytic constants and a moderate increase of the catalysis optimal pH (shift of 0.4 unit compared to WT). The R11A substitution also did not fully abolish the activity of the enzyme even though it decreased significantly (divided by a factor close to 250 as compared to WT). R11 formed a salt bridge with the N-terminal carboxylate group of GSH in the crystal structure (see above in Section 3.3). This interaction did not seem to be essential for the catalysis because the GSH apparent affinity in R11A was not much more degraded than in S10A (four and five times higher in S10A and R11A variants, respectively, compared to WT). The electrostatic influence of R11 on the catalytic process was, however, clear since the catalytic rate was 75 times lower in R11A compared to WT. This was accompanied by a significant one-unit increase in optimal pH suggesting a higher GSH-thiol pKa in the R11A variant than in the WT enzyme. These variations appeared small compared to those observed in eta GSTH1-1 from *Agrobacterium tumefaciens,* which harbored an arginine residue at the same position as in SynGSTC1. Indeed, the R34A mutation in AtuGSTH1-1 had a detrimental effect on the catalytic constant, which dropped by at least a factor of 5000 [58]. Finally, the substitution of S10 by a cysteine residue, had the same global effect as the S10A mutation (2.9 × 10^4^ M^−1^·s^−1^ for S10C vs. 3.6 × 10^4^ for S10A). Unlike the WT protein and other variants, the S10C enzyme was also active (*k*_cat_/*K*_m_ of 3.36 × 10^3^ ± 0.08 × 10^3^ M^−1^·s^−1^) with HED, a substrate commonly used to characterize Grxs and cysteinyl-GSTs, indicating that this variant acquired a significant thiol-transferase activity.

### 3.5. Comparative Genomic Analysis 

A similarity-based survey of 222 reference proteomes of the Cyanobacteria/Melainabacteria group led to the identification of 870 full-length GSTC1 homologues (BLASTP E-value cutoff 10^−3^). The phylogenetic analysis of these sequences led to a large tree (Appendix A). According to this tree, the glutathione transferase chi1 from *Synechocystis* sp. PCC 6803 (SynGSTC1) belonged to a large group of 147 sequences displaying a SRAS motif or related motifs (Appendix A and Appendix A). These 147 GSTC protein sequences displayed more than 35% of sequence identity and were largely distributed in Cyanobacteria, being present in 144 of the 208 analyzed cyanobacterial proteomes (Figure 5 and Appendix A for high-quality version). In contrast, they were absent in the noncyanobacterial members of the Cyanobacteria/Melainabacteria group. More precisely, they were present in all cyanobacterial orders excepted Gloeobacterales, the oldest branching extant group of cyanobacteria [59].

To go further, we inferred the phylogeny of the 147 sequences displaying the SRAS motif (or related motifs) (Appendix A). As expected, due to the restricted number of amino acid positions retained after the alignment trimming, branch supports were overall low (Appendix A). Despite this global lack of support, the resulting tree showed clearly that sequences harboring the SRAS motif and sequences harboring related motifs were mixed on the tree, indicating that the canonical SRAS motif was lost several times independently during the diversification of GSTCs. Furthermore, the topology of the tree also showed some inconsistencies with the phylogeny of species (Figure 5 and Appendix A). For instance, some Chroococcales sequences emerged within Nostocales (Appendix A), indicating that the evolutionary history of GSTCs harboring the SRAS (or related motifs) was impacted by horizontal gene transfers (HGTs) (Appendix A). Interestingly, these HGTs also contributed to spread GSTC1 outside of Cyanobacteria, since homologues were found in a few noncyanobacterial bacteria (Appendix A). Most of them were marine and freshwater bacteria and some were recently closely related to cyanobacteria as Planctomycetaceae bacterium TMED241. Indeed, it was found that this bacterium contains a circadian clock kaiABC operon, which is typically found in cyanobacteria [60].

The majority of the 147 GSTC sequences had a length of less than 190 amino acids (Appendix A). A dozen had longer sequences because they contained extensions at the N-terminus and/or between the secondary structures. All GSTCs had a reduced C-ter domain with a shortening of helices α4 and α5 as observed in the crystal structure of SynGSTC1. The SRAS motif (^10^SRAS^13^ in SynGSTC1) was well conserved. The arginine residue was invariant, the first position was replaced in a few cases by a threonine residue and the last two positions were a bit more variable. Surprisingly other residues involved in the structural attributes of SynGSTC1 were not conserved such as the patch of leucine residues (L70 and L94) in the core of the dimer, or the key residue of the lock and key motif (F49), or quaternary contributors to the stabilization of GSH (S98, T99, R116) (Figure 2). The sequence alignment revealed the conservation of 13 residues, most of which were located in the N-terminal domain (eight residues) and more precisely in the domain I (six residues) (Appendix A). The N-ter domain is generally better conserved than the C-ter domain because it contains an extended part of the active site [2]. In one subunit, the set of conserved residues was not centered on the active site but rather on the center of gravity of the monomer. The residues were distributed almost homogeneously around this center and most of them were located at a distance of less than 10 Å from it (Appendix A). This distribution was consistent with what is usually observed in proteins, namely that the most conserved positions tend to be situated in the core of the protein or on functional surfaces [61]. While the structural role of these conserved residues is obvious, it is difficult to identify those that form the signature of GSTs chi and most of them have be shown conserved in a class of GSTs. Only N97 seemed specific to the GST chi class; it most likely contributed electrostatically to the active site, as it was located near the γ-Glu moiety of GSH and close to the guanidinium group of the SRAS motif (Appendix A).

## 4. Conclusions

This study increased the knowledge on the biochemical characteristic acquired on the chi class of GSTs (GSTCs) and detailed for the first time the structural attributes of this GST class, specific to cyanobacteria. These short-sequence GSTs (~180 aa) had a three-dimensional structure with a very open active site because the α4 and α5 helices were significantly shorter than those usually observed. The glutathione substrate was tightly bound to the enzyme with its reactive center exposed to the solvent. The transfer of multipolar density parameters from small peptides to SynGSTC1 permitted the gradation of residues involved in GSH stabilization. The two carboxylate groups of GSH were the two chemical groups that best adhered to the protein. 

GSTCs contained a SRAS conserved motif at the N-terminus of the α1 helix indicating that they belonged to the SerGST group because the first residue of the motif was a serine residue. However, this serine residue was not directly involved in the catalytic act as assumed in SerGSTs [9]. The SRAS motif appeared to constrain the conformation of the serine side chain towards the interior of the protein and not towards the thiol group of GSH. S10 (in SynGSTC1) had a weak and unfavorable electrostatic influence on GSH and its mutation did not drastically alter the catalytic properties of the enzyme. The denomination TyrGST, CysGST, SerGST and AtyGST (tyrosine type GST, …, Atypical GST) has the advantage of simplifying the confusing and cumbersome Greek letter classification. It is relevant in the case of TyrGSTs from a phylogenetic point of view [1]. It is also appropriate in the case of CysGSTs because the cysteine residue is covalently bound to the substrate in one step of the catalytic mechanism [62]. The disadvantage of the residue-based naming is its stigmatization on one residue that may not have a strong link to the activity of the enzyme as is the case for SynGSTC1.

## Figures and Tables

**Figure 1 biomolecules-12-01466-f001:**
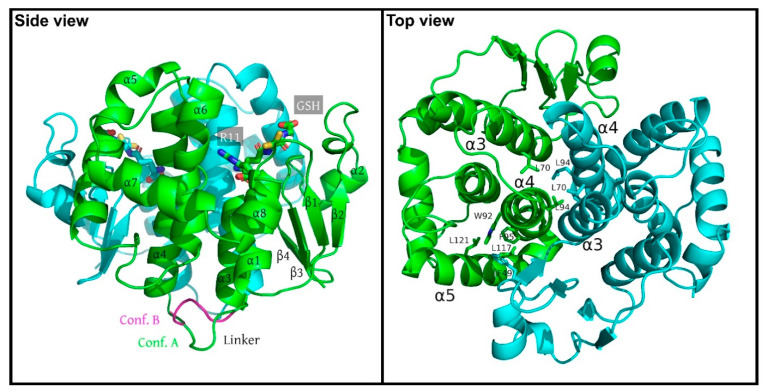
Crystal structure of the SynGSTC1 dimer, left, and rotated 90°, right. The monomers A and B are shown in ribbon mode and colored green and blue, respectively. (**Left**), side view. The secondary structures of the monomer A are labelled. In each monomer, the side chain of residue R11 and the glutathione molecules are labelled and highlighted as sticks. Both conformations of the linker are shown in monomer A. (**Right**), top view. The figure highlights the hydrophobic patches on SynGSTC1 dimer interface. L70 and L94 of both monomers are buried in the center of the dimer. This interaction pattern is complemented by a lock-and-key motif where the F49 residue (blue) fits into a low-polar cavity of the adjacent subunit (W92, F95, L117, L121) (green). The symmetry related lock-and-key motif is not shown for clarity.

**Figure 2 biomolecules-12-01466-f002:**
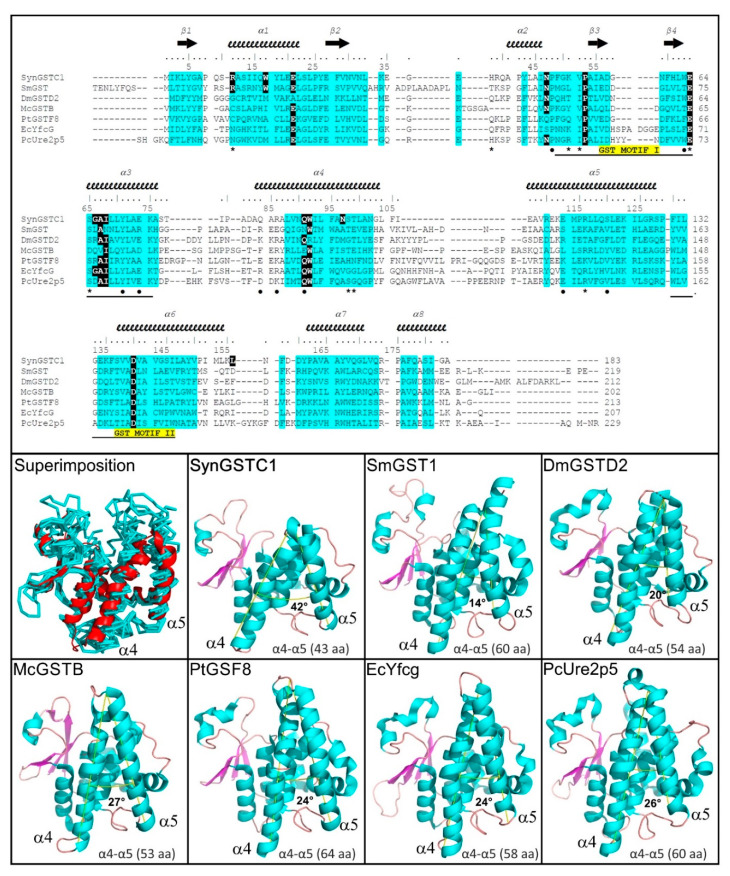
Comparison of SynGSTC1 with structural homologs. The top figure shows a structure-based sequence alignment, and the bottom figures highlight that SynGSTC1 has the shortest α4–α5 hairpin and the highest angle between α4 and α5 helices. Crystal structure and sequences can be found at the Protein Data Bank (http://www.rcsb.org, accessed on 30 September 2022): SynGSTC1, this study, PDB ENTRY 8AI8; SmGST, GST from *Sinorhizobium meliloti* 2011, PDB entry 4NHW; DmGSTD2, GST delta 2 from *Drosophila melanogaster*, PDB entry 5F0G; McGSTB, GST beta from *Methylococcus capsulatus* str. Bath, PDB entry 3UAP; PtGSTF8, GST phi 8 from *Populus trichocarpa*, PDB entry 5F07; EcYfcG, GST nu from *Escherichia coli* K-12, PDB entry 5HFK; PcUre2p5, Ure2p 5 from *Phanerodontia chrysosporium*, PDB entry 4F0C. The characteristics of the top figure are as follows: secondary structures are labelled and shown using arrows (β-strands) and squiggles (helices); common regions, i.e., regions with no gaps and with pairwise residue distances less than 4 Å are highlighted in blue; the invariant residues in the GST chi class are in bold type, coloured white and highlighted in black; residues that participates in dimer stabilization of SynGSTC1 via strong polar interactions are marked with ●; residues involved in binding glutathione (G-site) in SynGSTC1 are marked with *. The characteristics of the bottom figures are as follows: the models are shown in the cartoon or ribbon modes; the α4 and α5 helices are labelled; the first figure shows a superimposition of the seven structures where SynGSTC1 is colored red and the others cyan; in the other figures, the estimated angle between α4 and α5 helices is provided as well as the number of amino acids in the α4–α5 hairpin; the angles were calculated using the *AngleBetweenHelices* script (https://pymolwiki.org/index.php/AngleBetweenHelices, accessed on 30 September 2022) implemented in *PyMol Molecular Graphics System* (Version 2.0 Schrödinger, LLC, New York, NY, USA).

**Figure 3 biomolecules-12-01466-f003:**
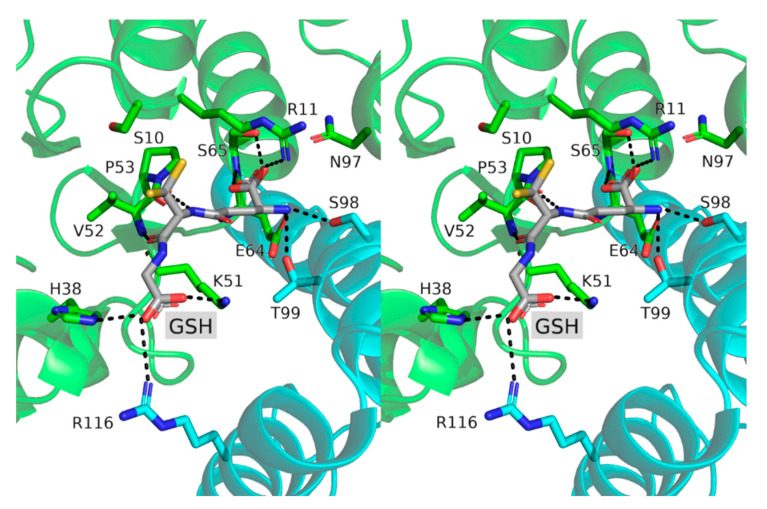
Stereoview of the glutathione binding site of SynGSTC1. The monomers A and B are shown in cartoon mode and colored green and blue, respectively. GSH and residues around it are shown as sticks and labelled. Numbering of residues is according to sequence of SynGSTC1. Strong intermolecular interactions are shown as dashed sticks.

**Figure 4 biomolecules-12-01466-f004:**
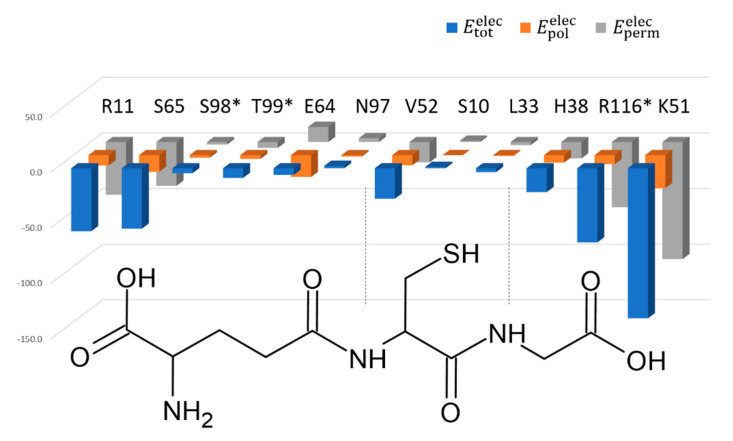
Permanent, polarization and total electrostatic interaction energies. The permanent $E_{\mathrm{perm}}^{\mathrm{elec}}$, polarization $E_{\mathrm{pol}}^{\mathrm{elec}}$ and total $E_{\mathrm{tot}}^{\mathrm{elec}}$ electrostatic interaction energies between the glutathione ligand and twelve residues of the SynGSTC1 active site are presented in kcal/mol. $E_{\mathrm{perm}}^{\mathrm{elec}}$ is computed using the electron density model transferred on the glutathione and the protein atoms, whereas $E_{tot}^{elec}$ is obtained after the electron density polarization procedure. Finally, $E_{\mathrm{pol}}^{\mathrm{elec}}$ is computed using $E_{\mathrm{pol}}^{\mathrm{elec}}=E_{tot}^{elec}-E_{\mathrm{perm}}^{\mathrm{elec}}$, and represents the polarization contribution to the total electrostatic interaction energy. The reported energy values have been averaged over the two conformations of the glutathione (A and B) observed in the crystal structure and over the two monomers. The GSH formula has been added to highlight the proximity of the residues to the GSH moieties. The residues marked with a star (*) in the figure are not from the same monomer as glutathione. The numerical values of these energies and the associated standard deviations are available in the Appendix A.

**Figure 5 biomolecules-12-01466-f005:**
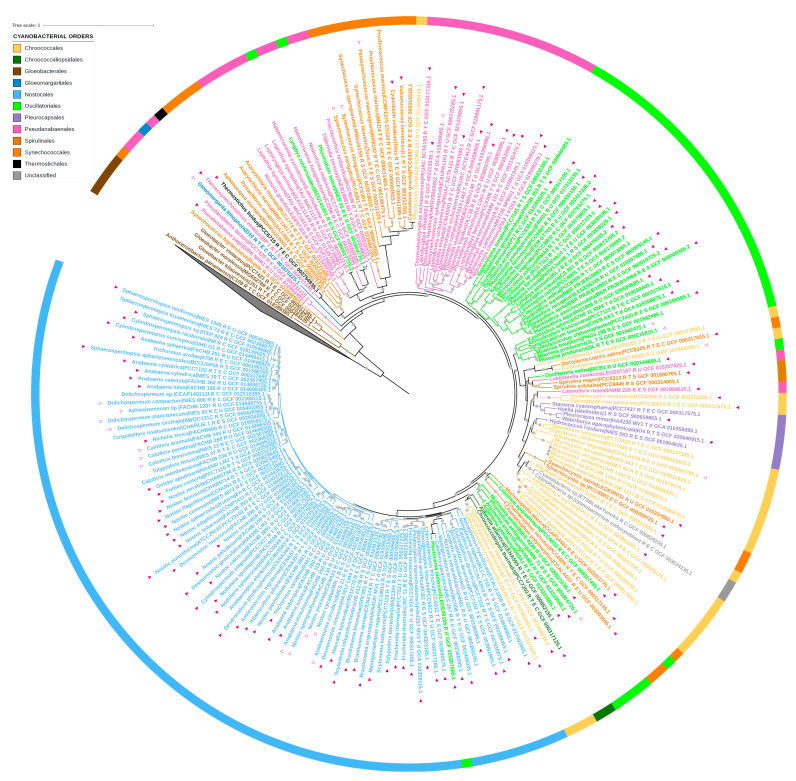
Phylogeny of the 222 proteomes of Cyanobacteria/Melainabacteria group considered in this study. The tree was inferred with *IQ-TREE* using the 52 rprots sequences present in more than 70% of the 222 proteomes (6430 amino acid sites, LG + C20 + F + R4 evolutionary model). The scale bar corresponds to the average number of substitutions per site. Gray circles correspond to ultrafast bootstrap values >90% (1000 replicates). The taxonomy of each proteome is indicated: Gloeobacterales (brown), Synechococcales (orange), Pseudanabaenales (pink), Gloeomargaritales (dark blue), Thermostichales (black), Oscillatoriales (light green), Chroococcales (yellow), Pleurocapsales (purple), Chroococcidiopsidales (dark green), Nostocales (light blue), and unclassified (gray). The 122 GSTC protein sequences harboring the SRAS motif are indicated with filled triangles, while the 25 GSTC sequences harboring variants of the SRAS motif are indicated with empty triangles. All the motifs are described in the Appendix A. The phylogeny of these 147 GSTC sequences is shown as Appendix A. A high-quality pdf version of the tree is provided as Appendix A in the online Appendix A.

**Table 1 biomolecules-12-01466-t001:** Statistics of X-ray diffraction data collection and model refinement.

	Wild-Type	S10T	R11A
Data Collection			
Diffraction source	ESRF-BM07	ESRF-BM07	ESRF-BM07
Detector	Pilatus 6M	Pilatus 6M	Pilatus 6M
Wavelength (Å)	0.97951	0.97951	0.97951
Space Group	*P*4_3_2_1_2	*P*4_3_2_1_2	*P*4_3_2_1_2
Unit-cell *a*; *c* (Å)	92.5; 193.6	92.9; 193.6	92.2; 193.1
Resolution Range (Å)	48.4 1.7 (1.73 1.70)	48.4 1.7 (1.73 1.70)	46.1 2.2 (2.27 2.20)
Tot. no. of meas. int.	1,200,217 (41158)	1,244,521 (62,785)	503,802 (22,268)
Unique reflections	92,986 (4529)	93,931 (4590)	39,298 (2006)
Average redundancy	13 (9)	13.2 (14)	13 (11)
Mean *I*/σ (*I*)	24.8 (1.8)	17.0 (2.0)	18.4 (2.4)
Completeness (%)	100.0 (99.6)	100.0 (100.0)	91.3 (55.5)
*R* _merge_	0.056 (1.097)	0.084 (1.52)	0.097 (1.039)
*R* _meas_	0.061 (1.168)	0.087 (1.59)	0.100 (1.142)
*CC* _1/2_	1.00 (0.83)	1.00 (0.84)	1.00 (0.85)
Wilson *B*-factor (Å^2^)	29.6	28.6	41.5
Refinement			
Resolution Range (Å)	24.8 1.7	24.2 1.7	31.3 2.2
No. of reflections	92839	93783	39248
*R*_work_/*R*_free_	0.204/0.221	0.206/0.221	0.215/0.242
Corr *Fo*-*Fc*/*Fo*-*Fc*_free_	0.938/0.936	0.940/0.939	0.907/0.888
Total number of atoms	3469	3500	3253
Average *B*-factor (Å^2^)	34.0	32.5	44.0
Model quality			
RMSZ Bond lengths	0.41	0.42	0.42
RMSZ Bond angles	0.54	0.56	0.56
Ramachandran fav. (%)	98	98	98
Ramachandran all. (%)	2	2	2
Rotamer outliers (%)	0	0	1
Clashscore	1	1	1
PDB entry	8AI8	8AI9	8AIB

*R*_merge_ = $\underset{hkl}{\sum }\underset{i}{\sum }\left|I_{i}\left(hkl\right)-I\left(hkl\right)\right|/\underset{hkl}{\sum }\underset{i}{\sum }I_{i}\left(hkl\right)$. *R*_meas_ = $\underset{hkl}{\sum }{\left\{N\left(hkl\right)/\left[N\left(hkl\right)-1\right]\right\}}^{1/2}\;\underset{i}{\sum }\left|I_{i}\left(hkl\right)-I\left(hkl\right)\right|/\underset{hkl}{\sum }\underset{i}{\sum }I_{i}\left(hkl\right)$. *CC*_1/2_ is the correlation coefficient of the mean intensities between two random half-sets of data. *R*_work_ = $\underset{hkl}{\sum }\left|\left|F_{obs}\right|-\left|F_{calc}\right|\right|/\underset{hkl}{\sum }\left|F_{obs}\right|$. In total, 5% of reflections were selected for *R*_free_ calculation. RMSZ: root mean square Z-score. The MolProbity clashscore is the number of serious clashes per 1000 atoms. Values in parentheses are for highest resolution shell.

**Table 2 biomolecules-12-01466-t002:** Kinetic parameters of SynGSTC1 toward model substrates.

	PITC	GSH	HED
***k*_cat_ (s^−1^)**			
WT	12.6 ± 0.2		ND
S10T	7.2 ± 0.1		ND
S10A	2.60 ± 0.05		ND
S10C	1.19 ± 0.02		0.0111 ± 0.0002
R11A	0.170 ± 0.003		ND
***K*_m_ (µM)**			
WT	33.8 ± 2.5	135.2 ± 7.9	ND
S10T	33.6 ± 2.6	142,8 ± 14.6	ND
S10A	89.7 ± 6.4	528.4 ± 33.0	ND
S10C	33.0 ± 3.0	2149 ± 123	3.3 ± 0.4
R11A	108.4 ± 6.0	719.2 ± 58.2	ND
***k*_cat_/*K*_m_ (M^−1^·s^−1^)**			
WT	3.73 × 10^5^ ± 0.06 × 10^5^		ND
S10T	2.14 × 10^5^ ± 0.04 × 10^5^		ND
S10A	2.89 × 10^4^ ± 0.06 × 10^4^		ND
S10C	3.60 × 10^4^ ± 0.06 × 10^4^		3.36 × 10^3^ ± 0.08 × 10^3^
R11A	1.53 × 10^3^ ± 0.02 × 10^3^		ND

The apparent *K*_m_ values of SynGSTC1 wild-type and variants (S10T, S10A, S10C and R11A) were determined by varying substrate concentrations at a fixed saturating GSH concentration. The apparent *K*_m_ and *k*_cat_ values were calculated with *Prism 8* software using the Michaelis–Menten equation as nonlinear regression model. Results are means ± S.D. (*n* = 3).

## Data Availability

PDB data (8AI8, 8AI9 and 8AIB) are made freely available by the wwPDB (https://www.wwpdb.org/ (accessed on 30 September 2022)).

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
