# Peer review of "Biochemical and Structural Characterization of Chi-Class Glutathione Transferases: A Snapshot on the Glutathione Transferase Encoded by sll0067 Gene in the Cyanobacterium Synechocystis sp. Strain PCC 6803"

_biomolecules, 2022, doi:10.3390/biom12101466_

Round 1

Reviewer 1 Report

The paper focuses on the biochemical characterization and the structure determination of a chi-class GST in the cyanobacterium Synechocystis sp.

The topic is interesting, and it increases the knowledge on bacterial GSTs.

The manuscript is well written, comprehensible, with meaningful value and it is suitable for Biomolecules.

This report would, however, be strengthened by some modifications, as detailed below.

- A preliminary crystallization of two GSTs from cyanobacteria were presented by Gun Stenberg and Mike Parker: “Crystallization and preliminary X-ray analysis of glutathione transferases from cyanobacteria by Feil et al. Acta Cryst. (2009). F65, 475–477, (doi:10.1107/S1744309109011634)”. Authors should insert the paper in the references and write a sentence in the introduction and conclusion sections.

- Pag.1, Abstract: reword “cyanobacteria” to “cyanobacterium”.

- Several typographic errors in the references probably due to the bibliography software used.

e.g.: reference n. 17:  spelling… Stenberg.

Author Response

Our response as been added as attached file.

Reviewer 2 Report

In this manuscript, the authors solve the structure of chi-class GST from cyanobacterium and do some enzymatic assays to analyze the important catalytic residue. However, there are some questions about the manuscript.

Some key questions:

1. Is that possible that SynGSTC1 reveals a novel class of GST superfamily, except TyrGST, CysGST, SerGST and AtyGST? The author needs to read more GST paper to analysis this classification.

2. In ref.18, it is shown that in chi class has the signature motif “GG[PA][KR]SRAS”, and also designates it as Y-type (TyrGST) because of the conserved residue Y5, Which is the same result in this manuscript. However, in the manuscript, the structure Y5 is far away from GSH. But if the author can provide some enzymatic assays about mutant of Y5, to show no effect of the activity, then it means that SynGSTC1 does not belong to TyrGST. It is the same as the author indicates that SynGSTC1 does not belong to SerGST.

3. In the manuscript the authors also show that R11A decreases 250 times of the enzyme activity compared with the WT, and SRAS is the active site motif, how about other SerGST, it is classified as the SerGST family because the first residue of this motif or Ser is the most important residues?

4. Show the structure figure of this comparison. “α4-α5 hairpin pattern is significantly shortened (roughly 10 residues) and the angle between these two helices (~30°) is twice that usually observed. This "missing" region makes the active site of SynGSTC1 very open, with no clear pocket for the hydrophobic substrate (H-site)” It seems this is also an essential finding of this manuscript.

Additional questions & comments:
1. In the method part, after the addition of 10 mM imidazole, SynGSTC1 was eluted from Ni-NTA. Usually, for the His-tag protein purification, the elution buffer needs to have more than 50 mM imidazole.

2. Fig.S1 is beautiful, and it is good to move Fig. S1 to Fig.1. The authors can make more panels on each figure.

3. In Fig.1 and Fig.3 legends, the structures are shown as cartoon mode, not ribbon mode.

4. In Fig.2, representation and protein sequence are not aligned, like residues involved in binding glutathione (G-site) in SynGSTC1 are marked with *, but in the figure, some of the * indicate the wrong position.

5. Which structure shows that the equivalent serine residue interacts with GSH?  Compare the two structures and sequence alignment of these sequences to confirm that S10 is conserved and important for catalysis.

6. What is the distance between S10 to GSH?

7. Delete this sentence “In any case, the classification of GSTs based on a single residue appears misleading and subject to generalizations resulting in false predictions”. Re-written the conclusion part.

8. The manuscript is very hard to follow and should be considerably re-written.

 In the abstract “participate to the detoxification process” should be “participate in the detoxification process”. The tense in the abstract should be consistent.

In the introduction, “according the enzyme mechanism” should be “according to the enzyme mechanism”. “The important residue of the others (TyrGSTs, SerGSTs and AtyGSTs) has a spectator role enhancing the nucleophilicity of the glutathione thiolate group, its mutation very often does not abolish the activity” can revise as “The important residue of the others (TyrGSTs, SerGSTs and AtyGSTs) has a spectator role in enhancing the nucleophilicity of the glutathione thiolate group, and the mutation of this residue does not abolish the activity”. “studies in bacteria remain scarce” should be “studies on bacteria remain scarce”

“Other classes exist to which no Greek letter has been given and which have specific functions such as LigE, LigF and LigG involved in lignin degradation in soil bacteria” needs to be re-written.

“combined to” should be “combine with”
If SynGSTC1 (Sll0067) is used as a gene name, it needs to revise. Gene symbols are italicized, first letter upper case all the rest lower case.

And many confusing writings and mistakes in the results part.

Author Response

(The authors gave the same response as above.)

Reviewer 3 Report

This manuscript presents a very thorough structural, enzymatic and phylogenetic analysis of the chi-class GST from Synechocystis sp. PCC 6803.

(Arguably) the most interesting aspect of the paper is the finding that there is no "critical" catalytic residue for activating the thiol group of glutathione (as supported by the structural and mutagenesis studies).

The experimental aspects appear sound.

Corrections:

The main body of the text could use a figure showing SynGSTC1 with its structural neighbours (such as those used to generate figure 2), either overlayed or side-by-side in the same orientation, to help orient readers.

The quantification of electrostatic interaction energies appears to be a useful new tool for objectively assessing interaction energies of residues with ligands. With respect to the interaction of SynGSTC1 with glutathione, it would be useful for readers unfamiliar with the technique a similar analysis of a GST for which a key serine or tyrosine residue have been confirmed to play an important role in catalysis. (e.g. Lucilia cuprina GST).

Figure 2: Should not be wrapped across pages. Check the secondary structure indicators. They appear to be out of alignment. Similarly the star and circle markers appear to be out of alignment with the sequences.

Figures 5, S10 and S11: the text is unreadable! Even if the resolution was higher in the final version, it is really going to strain the eyesight Would it be more useful to show just the relationship between major orders?

p11: exponents of powers of 10 not superscript in "Unlike the WT protein and other variants, the S10C enzyme was also active (kcat/Km of 3.36x103 ± 0.08x103 M-1.s-1)"

Author Response

(The authors gave the same response as above.)

Round 2

Reviewer 2 Report

The authors have addressed all of my concerns.